# Laboratory Performance Evaluation of a Low-Cost Electrochemical Formaldehyde Sensor

**DOI:** 10.3390/s23177444

**Published:** 2023-08-26

**Authors:** Zheyuan Pei, Maxim Balitskiy, Ryan Thalman, Kerry E. Kelly

**Affiliations:** 1Department of Chemical Engineering, University of Utah, Salt Lake City, UT 84112, USA; zheyuan.pei@utah.edu (Z.P.); maxim.balitskiy@utah.edu (M.B.); 2Department of Chemistry, Snow College, Ephraim, UT 84627, USA; ryan.thalman@snow.edu

**Keywords:** formaldehyde, low-cost electrochemical sensor, broadband cavity-enhanced absorption spectroscopy, sensor evaluation

## Abstract

Formaldehyde is a known human carcinogen and an important indoor and outdoor air pollutant. However, current strategies for formaldehyde measurement, such as chromatographic and optical techniques, are expensive and labor intensive. Low-cost gas sensors have been emerging to provide effective measurement of air pollutants. In this study, we evaluated eight low-cost electrochemical formaldehyde sensors (SFA30, Sensirion^®^, Staefa, Switzerland) in the laboratory with a broadband cavity-enhanced absorption spectroscopy as the reference instrument. As a group, the sensors exhibited good linearity of response (*R^2^* > 0.95), low limit of detection (11.3 ± 2.07 ppb), good accuracy (3.96 ± 0.33 ppb and 6.2 ± 0.3% *N*), acceptable repeatability (3.46% averaged coefficient of variation), reasonably fast response (131–439 s) and moderate inter-sensor variability (0.551 intraclass correlation coefficient) over the formaldehyde concentration range of 0–76 ppb. We also systematically investigated the effects of temperature and relative humidity on sensor response, and the results showed that formaldehyde concentration was the most important contributor to sensor response, followed by temperature, and relative humidity. The results suggest the feasibility of using this low-cost electrochemical sensor to measure formaldehyde concentrations at relevant concentration ranges in indoor and outdoor environments.

## 1. Introduction

Globally, formaldehyde is the most abundant carbonyl [1]. It is also a known human carcinogen [2,3]. More than 25 million people in the US are exposed to formaldehyde levels that exceed the cancer risk threshold, making it the biggest driver of cancer risk among hazardous air pollutants [4]. Hazardous air pollutants are defined by the US EPA as compounds that known to cause cancer and other serious health impacts [5]. Formaldehyde also plays a key role in ozone and secondary particulate matter formation—both of which are associated with significant adverse health effects [6,7,8]. Should formaldehyde sources be better characterized and controlled, it will reduce health risks to the local citizens and help address critical air pollution challenges: PM_2_._5_ and ozone.

The existing measurement techniques for formaldehyde can be broadly categorized as satellite-based, spectrometric, and optical techniques. Satellite-based estimates of average annual formaldehyde concentration have been made at a grid resolution of approximately 5 × 5 km^2^ [9]. This resolution is excellent for monitoring regional and global trends, but is unsuitable for identifying local emission sources. Chromatographic technologies, such as ion mobility spectrometry (IMS) and gas chromatography-mass spectrometry (GC-MS) are gold standards for aldehyde identification [10]. In practice, samples are usually collected in canisters, which are analyzed by contract labs at a high cost per sample and results can be delayed for weeks to months. Although portable systems exist, they tend to be prohibitively expensive. Optical techniques include differential optical absorption spectroscopy (DOAS), infrared spectrophotometry, Raman spectroscopy, fluorescence, and colorimetric [1,11]. These methods provide sensitive measurements for a wide range of VOCs, but they are labor-intensive and require daily maintenance. Overall, these strategies for formaldehyde measurement are either expensive or labor-intensive, which hamper the ability to identify formaldehyde sources.

Low-cost gas sensors have been emerging in recent years, which can complement measurements from regulatory or research-grade measurements [12,13,14,15,16,17,18,19]. These sensors can be easy to deploy and provide real-time measurement with higher spatial coverage than conventional measurements [18]. Common low-cost formaldehyde sensors can be categorized as semiconductor sensors and electrochemical sensors [16,18,20]. Semiconductor sensors can detect formaldehyde by enabling a redox reaction and then measuring the resistance change [21]. These sensors have high sensitivity and stability. However, some of them, such as metal oxide sensors, require high operating temperature (>100 °C) and have low selectivity [20,21,22]. On the other hand, electrochemical sensors detect formaldehyde by enabling an electrochemical reaction and then measuring the current change [23]. These sensors can have high selectivity and accuracy. In addition, they consume less power compared to metal oxide sensors because the sensors generally work under room or ambient temperature. However, the signals of electrochemical sensors can be influenced by temperature and relative humidity (RH) [24,25].

Some laboratory studies have evaluated the performance of low-cost formaldehyde sensors. Chattopadhyay et al. [26] evaluated the performance of two electrochemical sensors and three metal oxide sensors in a laboratory chamber over the concentration range of 10–800 ppb, a temperature range of 22–50 °C, and a RH range of 8–85%. Deng et al. [27] developed a metal oxide sensor based on hierarchical flower-like CuO nanostructure and evaluated the sensor response to a formaldehyde concentration range of 50–1000 ppb. Gautam et al. [28] developed a Si-chip assisted MOS/SiNWs nanocomposite-based sensor and evaluated the sensor response to a formaldehyde concentration range of 0.01–1000 ppm. Li et al. [29] synthesized SnO_2_ microspheres and evaluated the sensor response to a formaldehyde concentration range of 1–500 ppm. Hu et al. [30] reported batch fabrication of formaldehyde sensors based on LaFeO_3_ film and evaluated sensor response over a concentration range of 0.05–1 ppm. However, these studies have some limitations. First, most of the studies focused on sensor fabrication and provided limited sensor performance metrics. Most of the studies reported the linearity of response [28,29,30], but only a few studies reported limit of detection (*LOD*) [27,30], sensor accuracy [26], or sensor repeatability [28,29]. Second, most of the studies focused on formaldehyde concentration ranges that were higher than typical indoor (~17 ppb) [31] or ambient (~3 ppb) [32] levels in the United States. In addition, previous studies have reported the influence of temperature and RH on the performance of electrochemical sensors [25,33,34,35,36]. However, some studies report contradictory results regarding the effect of RH [37], and many studies only report qualitative results [30,38,39,40]. The effects of temperature and RH on electrochemical formaldehyde sensors have not been systematically investigated, which is crucial to understanding sensor performance in the ambient environment.

This study aims to evaluate the laboratory performance of eight low-cost electrochemical formaldehyde sensors (Sensirion SFA30^®^, Staefa, Switzerland) over an environmentally relevant concentration range of 0–76 ppb, compared to a high-accuracy broadband cavity enhanced absorption spectrometer (BBCEAS) as the reference instrument [41]. It provides comprehensive performance metrics including linearity of response, *LOD*, accuracy, repeatability, response time, and inter-sensor variability. This study also aims to systematically investigate the effects of temperature and RH on sensor performance by implementing a Box-Behnken experimental design and developing a multiple linear regression (MLR) model. Understanding the sensor performance in a laboratory environment paves the way to future studies of applying the sensors to field measurement of formaldehyde.

## 2. Materials and Methods

### 2.1. Sensor Preparation

This study evaluated eight Sensirion SFA30^®^ formaldehyde sensors (Sensirion AG, Staefa, Switzerland). These sensors had integrated temperature and RH modules. The laboratory evaluation entailed exposing the low-cost sensors to target concentrations of formaldehyde, temperature, and RH. Each sensor was placed in a 3D-printed polylactic acid (PLA) chamber, which was sealed with polytetrafluoroethylene (PTFE) tape to minimize leakage. These eight sensors were divided into two sets. Each set included four sensors that were connected to an Arduino board via a multiplexer (I^2^C interface), as shown in Appendix A. This setup allowed real-time viewing and recording of the sensor signal, temperature, and RH using the Arduino interface on a computer.

### 2.2. Broadband Cavity-Enhanced Absorption Spectrometer

A BBCEAS provided the reference formaldehyde measurements [42] for this study. The BBCEAS consisted of a light source module, an optical cavity module, and a detection module, as shown in Appendix A. A deep ultraviolet LED (DUV-325, Roithner LaserTechnik, Vienna, Austria) centered at 325 nm provided the light source. The LED was aligned within a PTFE optical cavity, which contained two highly reflective mirrors (Layertec GmbH, Mellingen, Germany). An optical fiber collected light at the rear of the cavity, and this optical fiber was connected to a high-sensitivity optic spectrometer (AvaSpec-Hero, Avantes, Inc., Apeldoorn, Netherlands ). Spectra were saved every minute.

The BBCEAS measurements were given by:(1)$$\alpha \left(\lambda \right)=\left[\frac{1-R\left(\lambda \right)}{d}+{\alpha \left(\lambda \right)}_{Rayleigh,ZA}\right]\left[\frac{I_{ZA}\left(\lambda \right)-I\left(\lambda \right)}{I\left(\lambda \right)}\right]$$
where $\alpha \left(\lambda \right)$ is the extinction coefficient of the transmitted light through the cavity, λ is the wavelength of light, d is the cavity length, $R\left(\lambda \right)$ is the mirror reflectivity, ${\alpha \left(\lambda \right)}_{Rayleigh,ZA}$ is the extinction due to Rayleigh scattering of zero air, $I_{ZA}\left(\lambda \right)$ is the reference spectrum of zero air, and $I\left(\lambda \right)$ is the measured spectrum at each wavelength.

We obtained the mirror reflectivity by flowing 1 LPM of nitrogen for 10 min followed by flowing 1 LMP of helium for 10 min into the BBCEAS. Mirror reflectivity was obtained using the following relationship:(2)$$R\left(\lambda \right)=1-d\cdot \frac{I_{N_{2}}\cdot n_{N_{2}}\cdot \sigma_{Ray,N_{2}}\left(\lambda \right)-I_{He}\cdot n_{He}\cdot \sigma_{Ray,He}\left(\lambda \right)}{I_{He}\left(\lambda \right)-I_{N_{2}}\left(\lambda \right)}$$
where d is the cavity length, $I_{N_{2}}\left(\lambda \right)$ and $I_{He}\left(\lambda \right)$ are the measured spectral intensities when the cavity is filled with nitrogen and helium, respectively, $\sigma_{Ray,N_{2}}\left(\lambda \right)$ and $\sigma_{Ray,He}\left(\lambda \right)$ are the Rayleigh scattering cross section of nitrogen and helium, respectively.

Subsequently, we obtained the reference spectra by flowing 1 LPM of zero air into the BBCEAS for 20 min. For each formaldehyde sample gas, the sampling time was 75 min. Extinctions from the cavity were fitted using standard literature absorption cross-sections for formaldehyde [43]. All spectra were post-processed using Equations (1) and (2) with a custom-made Python package to obtain the formaldehyde concentration. To obtain formaldehyde concentration from the BBCEAS, we averaged 30 min of stable spectra for the concentration-only tests and 10 min of stable spectra for the environmental simulation tests.

### 2.3. Laboratory Evaluation System

We designed a laboratory system to evaluate four formaldehyde sensors simultaneously in parallel (Figure 1). The system contained two gas lines, a blank line, and a sample line. Both the blank and sample lines were set to 1 LPM using a mass flow controller (MFC, Cole Parmer, Inc., Vernon Hills, IL, USA). The blank line consisted of zero air, which provided the baseline signal for both the sensors and the BBCEAS. This blank line was also used to flush the system. The sample line started from the permeation oven (Dynacalibrator^®^ Model 340, VICI, Inc., Houston, TX, USA), which generated formaldehyde with a permeation tube of paraformaldehyde (325 ng/min, VICI, Inc., Houston, TX, USA) heated to 90 °C. By adjusting the dilution rate of the permeation oven, formaldehyde concentrations ranging from 0–76 ppb were obtained. A switching valve (Parker, Inc., Mayfield Heights, OH, USA) allowed the two lines to alternate.

The evaluation system was capable of generating different temperature and RH conditions with its environmental chamber, which consisted of a temperature module (water bath, Precision, Thermo Fisher Scientific, Waltham, MA, USA) and an RH module (Nafion tube, Perma Pure LLC, Lakewood, NJ, USA), which provided a temperature range of 0–40 °C and an RH range of 15–75%. The water bath provided the heat source or sink for the environmental chamber. A 6” Nafion (OD 0.110”) tube immersed in deionized water inside a closed glass container provided humidity. A humidity bypass was implemented to adjust the dry/humid gas ratio and provide the target humidity. The sample gas with target concentration, temperature, and RH flowed past the four sensors and then entered the BBCEAS.

For all tests in this study, we used 75 min as the sampling time to obtain relatively stable formaldehyde concentration, sensor signal, temperature, and RH. In between any two samples, we flushed the system with zero air for 20 min. The sensor signal, temperature, and RH from the Arduino interface were recorded and post-processed by taking the average of each stable signal (30-min average for the concentration-only tests and 10-min average for the environmental simulation tests).

### 2.4. Experimental Design

We evaluated the sensor response to formaldehyde concentration alone (concentration-only tests) and their response to different formaldehyde concentrations, temperature, and RH conditions (environmental simulation tests).

Concentration-only tests

The concentration-only tests included target formaldehyde concentrations of 10, 20, 30, 40, and 50 ppb at 24 °C and approximately 15% RH.

b.Effects of temperature and RH

To further investigate the sensor performance under different temperature and RH conditions, we performed environmental simulation tests with a Box-Behnken experimental design (Table 1). This Box-Behnken design effectively reduced the total number of tests needed to evaluate the influence of temperature and RH on sensor response to formaldehyde.

### 2.5. Data Analysis

We applied ordinary least squares (OLS) to evaluate the effects of the different parameters on sensor response. All eight sensors were evaluated both individually and as a group.

Linear regression (LR) model

For the concentration-only tests, a linear regression (LR) model provided the relationship between the sensor response and the reference measurements (BBCEAS):(3)$$\hat{y}=kc+b$$
where c is the formaldehyde concentration from the BBCEAS, k is the slope of the linear regression, b is the intercept of the linear regression and $\hat{y}$ is the predicted sensor response based on the LR model.

b.Multiple linear regression model

An MLR model provided the relationship between the sensor response to formaldehyde concentration, temperature, and RH:(4)$$\hat{y}=k_{1}c+k_{2}T+k_{3}RH+b$$
where $\hat{y}$ is the predicted sensor signal, c is the formaldehyde concentration from the BBCEAS, T is the temperature, RH is the relative humidity, b is the intercept, and $k_{1}$, $k_{2}$, $k_{3}$ are the coefficients of the environmental factors.

### 2.6. Sensor Performance Metrics

Although the performance guidelines for low-cost formaldehyde sensors do not exist, we discussed typical performance metrics (linearity of response, *LOD*, accuracy, and repeatability), which have been used to evaluate other low-cost air quality sensors [16,18]. In addition, the U.S. environmental protection agency (EPA) also uses these metrics for the evaluation of low-cost gas sensors [44]. We also investigated into the response time and inter-sensor variability.

Linearity of response

In this study, we used the coefficient of determination (*R^2^*) to evaluate linearity of response:(5)$$R^{2}=1-\frac{\sum {\left(y_{i}-{\hat{y}}_{i}\right)}^{2}}{\sum {\left(y_{i}-\bar{y}\right)}^{2}}$$
where $y_{i}$ is the observed sensor signal, $\bar{y}$ is the mean of the observed sensor signal, ${\hat{y}}_{i}$ is the predicted sensor signal.

b.Limit of detection

The *LOD* was calculated based on the linear relationship between each sensor and the BBCEAS [45]:(6)$$LOD=\frac{3.3\sigma }{k}$$
where σ is the standard error of the linear regression and k is the slope of the linear regression from Equation (3).

c.Sensor accuracy

We used root mean squared error (*RMSE*) and normalized root mean squared error (*NRMSE*) to evaluate sensor accuracy:(7)$$RMSE=\sqrt{\frac{1}{N}\sum_{i=1}^{N}{\left({\hat{y}}_{i}-c_{i}\right)}^{2}}$$
(8)$$NRMSE=\frac{RMSE}{c_{max}-c_{min}}$$
where ${\hat{y}}_{i}$ is the predicted sensor signal, $c_{i}$ is the observed formaldehyde concentration from the BBCEAS, $c_{max}$ is the maximum of the observed formaldehyde concentration and $c_{min}$ is the minimum of the observed formaldehyde concentration, N is the total number of measurements.

In addition to, some other studies of formaldehyde sensor performance have presented mean absolute error (*MAE*) [26]. *MAE* is given by:(9)$$MAE=\frac{\sum \left|{\hat{y}}_{i}-c_{i}\right|}{N}$$
where ${\hat{y}}_{i}$ is the predicted sensor signal, $c_{i}$ is the observed formaldehyde concentration from the BBCEAS, N is the total number of measurements.

d.Sensor repeatability

To evaluate the ability of the sensors to generate reproducible responses to the same formaldehyde concentrations, the concentration-only tests were repeated three times. Coefficient of variation (*CV*) is a common metric for low-cost sensor repeatability [16], and it is given by the following equation:(10)$$CV=\frac{\sigma }{\mu }\times 100\%$$
where σ is the standard deviation of repeated measurements and μ is the mean of repeated measurements.

However, in this study, the target formaldehyde concentrations could not be reproducibly achieved because of two reasons. First, the dilution rate of the permeation oven could not be precisely controlled. Secondly, the BBCEAS provided formaldehyde concentration only after post-processing at the end of each test; hence the formaldehyde concentration could not be dynamically adjusted to the target concentration. Consequently, we conservatively estimated the *CV* by calculating the mean and standard deviation of the sensor responses for each target formaldehyde concentration.

e.Response time

The response time of the sensors was estimated by *t*_90_, which is the time needed for the sensors to reach 90% of the final stable signal [46].

f.Inter-sensor variability

We implemented intraclass correlation coefficient (*ICC*) to examine inter-sensor variability [47]. Widely used in reliability testing, *ICC* is a number ranging from 0 to 1, which refers to the correlations within a class of data [48]. Based on the 95% confidence interval of the *ICC* estimate, values less than 0.5, between 0.5 and 0.75, between 0.75 and 0.9, and greater than 0.90 are indicative of poor, moderate, good, and excellent reliability, respectively [49]. In this study, we used a linear mixed effect model implemented in R Studio^®^ (version 4.2.2) to calculate the *ICC*, where formaldehyde concentration was the fixed effect and the sensor IDs were the random effect:(11)$$y=kc+b+b_{s}+\varepsilon$$
where y is the sensor signal, c is the formaldehyde concentration from the BBCEAS, k is the slope of the linear regression, b is the intercept of the linear regression, $b_{s}$ is the random intercept that captures the variability of sensor IDs, ε is the random error.

Consequently, *ICC* was calculated as:(12)$$ICC=\frac{{\sigma_{b_{s}}}^{2}}{{\sigma_{b_{s}}}^{2}+{\sigma_{\varepsilon }}^{2}}$$
where ${\sigma_{b_{s}}}^{2}$ is the variance of the random effect and ${\sigma_{\varepsilon }}^{2}$ is the variance of the random error.

### 2.7. Preliminary Cross-Sensitivity Tests

Although electrochemical sensors generally have high selectivity, previous studies have reported cross sensitivity to common atmospheric trace gases [12,25,28,50,51]. In this study, we considered common ambient trace gases, including carbon monoxide (CO), nitric oxide (NO), nitrogen dioxide (NO_2_) and isobutylene. The tests were performed using the same experimental setup as the concentration-only tests, except for the gas generation and reference instrument. Four sensors were exposed to each target gas at the following concentrations: CO (39.7 ppm), NO (101 ppb), NO_2_ (83 ppb), and isobutylene (100 ppb). The target CO, NO, and NO_2_ concentrations were based on US National Ambient Air Quality Standards [52]. Each target gas concentration was obtained by diluting a calibration gas cylinder with zero air. The concentrations of CO and isobutylene were measured by a Q-Trak 7575-X (TSI Inc., Shoreview, MN, USA). The concentrations of NO and NO_2_ were measured by Model 42i NO-NO_2_-NO_X_ Analyzer (Thermo Fisher Scientific Inc., Waltham, MA, USA). The tests were performed at a temperature of 24 °C and a RH of approximately 15%. We compared sensors signals between zero air and each target gas individually using a Student’s *t*-test.

## 3. Results and Discussion

### 3.1. Concentration-Only Tests

Figure 2 shows the response of the eight sensors to formaldehyde concentrations ranging from 10.7 ppb to 70.9 ppb (at 24 °C and approximately 15% RH). Appendix A shows the sensor responses for the other two repeated tests. In general, the sensor response increased linearly as formaldehyde concentration increased. Previous laboratory studies of formaldehyde sensors found a linear relationship between sensor response and formaldehyde concentration [29,30,53]. These previous studies included a LaFeO_3_ thin film sensor, a SiO_2_ microsphere sensor, and a Ni-Doped SnO_2_ nanoparticle sensor. In this study, a linear regression model was applied to the concentration-only tests. As shown in Appendix A, the slopes of the three repeated concentration-only tests are 1.11 ± 0.097, 1.13 ± 0.086, and 1.07 ± 0.090. The intercepts of the three repeated tests are −7.39 ± 2.44, −8.98 ± 2.02 and −6.68 ± 2.93 ppb.

We evaluated the performance of the formaldehyde sensors both as a group and individually (Section 2.6 describes the performance metrics). Table 2 shows the performance metrics for the eight sensors evaluated as a group. Figure 3 and Appendix A show the performance metrics for individual sensors.

Linearity of response

As shown in Figure 2 and Figure 3A, the sensors demonstrated an *R^2^* > 0.98 for individual sensors and an *R^2^* > 0.95 as a group within the target formaldehyde concentration range of 0–50 ppb. Linearity of formaldehyde sensor response has been widely reported in previous laboratory studies, and linear regression models have often been applied to evaluate formaldehyde sensors compared to research-grade instrumentation. Alonso et al. [37] reported an *R^2^* = 0.775 using OLS in their evaluation of low-cost micro fuel cell formaldehyde sensors over the concentration range of 17.3–477.1 ppb (22–606 µg/m^3^). Hu et al. [30] fabricated a LaFeO_3_ thin-film formaldehyde sensor and reported an *R^2^* = 0.993 over the concentration range of 0.05–1 ppm in a laboratory test. An *R^2^* value greater than 0.75 is generally considered as a strong agreement between a low-cost sensor and a reference instrument [16,18]. Therefore, our sensors demonstrate a good linearity of response.

b.Limit of detection

In this study, the *LOD* (Figure 3B) was calculated based on the linear regression of the sensor response vs. the BBCEAS. Our results show that the individual *LOD* of the sensors ranged from 2.86–9.73 ppb. As a group, the sensors demonstrated an *LOD* of 11.3 ± 2.07 ppb. Overall, the sensors exhibit an *LOD* that is lower than those reported in previous laboratory studies. Alolaywi et al. [38] evaluated an electrochemical formaldehyde sensor and reported a *LOD* of 60 ppb. Evaluations of nanomaterial-based formaldehyde sensors have reported *LOD*s ranging from 50–90 ppb [27,40,53]. It is important for low-cost sensors to be capable of measuring relevant levels of formaldehyde. Hun et al. [31] reported a mean indoor formaldehyde concentration of 17 ppb. Typical ambient formaldehyde concentrations in urban areas of the U.S. range from 11–20 ppb [54] with some industrial areas of the U.S. reporting higher levels of formaldehyde (24–66 ppb) [55]. Our low *LOD* suggests the potential of using the sensors to measure indoor and ambient levels formaldehyde.

c.Sensor accuracy

The accuracy of the sensors was characterized by *RMSE* (Figure 3C) and *NRMSE* (Figure 3D). In the three repeated concentration-only tests, the individual *RMSE* ranged from 1.84 ppb to 10.1 ppb, and the individual *NRMSE* ranged from 2.9% to 16.8%. Sensor S6 and S8 exhibited higher *RMSE*s (8.76 ± 1.01 ppb and 7.22 ± 1.95 ppb, respectively) compared to the rest of the sensors. As a group, the eight sensors demonstrated an *RMSE* of 3.96 ± 0.33 and an *NRMSE* of 6.23 ± 0.33%, indicating low errors between the measurements of the low-cost sensors and the reference instrument. Although accuracy criteria for low-cost formaldehyde sensors do not exist, for discussion purposes, we compared the *RMSE* and *NRMSE* values of the formaldehyde sensors to EPA’s criteria for low-cost ozone sensors (*RMSE* ≤ 5 ppb) [56] and particulate matter sensors (*NRMSE* ≤ 30%) [57]. These comparisons suggest that the sensors evaluated in this study would meet EPA guidelines for low-cost sensor accuracy.

Limited laboratory studies have reported *RMSE* or *NRMSE* for formaldehyde sensors. Song et al. [58] developed an yttrium-doped ZnO sensor array with a back propagation neural network model, reporting an *RMSE* of 0.892 ppm over a formaldehyde concentration range of 5–45 ppm in a chamber test. In this study, the individual *MAE* of the sensors in three repeated tests ranged from 1.53 ppb to 10.1 ppb, as shown in Appendix A. Sensor S6 and S8 demonstrated higher *MAE*s (8.66 ± 1.02 ppb and 7.04 ± 1.82 ppb, respectively) compared to the rest of the sensors in the three repeated concentration-only tests. Chattopadhyay et al. [26] evaluated both electrochemical (*MAE* ranging from 70.8 ppb to 78.8 ppb) and metal oxide (*MAE* ranging from 154 ppb to 335 ppb) formaldehyde sensors over a concentration range of 10–800 ppb in a laboratory environment. Our results demonstrated lower measurement error between the low-cost sensors and the reference instrument.

d.Sensor repeatability

In addition to accuracy, repeatability is another important factor in the evaluation of low-cost sensors. As discussed in Section 2.6, we conservatively estimated the *CV* of the eight sensors, which ranged from 1.36 to 6.41% (Appendix A). Some previous studies [28,29,59] reported that the formaldehyde sensors demonstrated good repeatability by repeatedly exposing the sensors to the same formaldehyde concentration. However, these studies lack quantitative evaluation of the repeatability. Alonso et al. [37] reported a *CV* of 20%, 14.86% and 14.82% for three different models of a micro fuel cell formaldehyde sensor. Although *CV* criteria for low-cost formaldehyde sensors do not exist, EPA recommends *CV* ≤ 30% for both ozone [56] and PM_2_._5_ [57] sensors. Our conservative estimation indicates that the low-cost formaldehyde sensors have acceptable repeatability.

e.Response time

Appendix A summarizes the response time for sensor S1-S4 in the concentration-only tests. In general, the response time increased as the formaldehyde concentration increased. We observed a minimum response time of 131 s and a maximum response time of 439 s over the formaldehyde concentration range of 17.2–60.1 ppb. Our sensors demonstrate comparable response times to previous studies. Descamps et al. [39] evaluated a hand-held fluorescence-based sensor in the laboratory and reported a response time of 180 s for formaldehyde concentrations ranging from 0–30 ppb. Evaluations of thin-film semiconductor sensors presented response times of less than 120 s for ppm-level formaldehyde concentrations [60].

f.Inter-sensor variability

Although the eight sensors were purchased from the same distributor at the same time, some inter-sensor variability still existed. As a result, even though the sensors were exposed to the same formaldehyde concentration, they exhibited slightly different responses, which were reflected in their slopes, intercepts, *R^2^* values, *LOD*s, *RMSE*s, *NRMSE*s and *MAE*s (Appendix A). This study also evaluated inter-sensor variability using the ICC. Sensors S1-S8 showed an ICC of 0.551, which corresponds to moderate variability (ICC: 0.50–0.75) [49].

### 3.2. Environmental Simulation Tests

We evaluated the effects of temperature and RH on sensor response and developed an MLR model for sensor response (Figure 4 and Appendix A). MLR is a commonly used model to estimate gas concentrations from the low-cost sensors, and common parameters include target gas concentration, temperature, RH and interferent gases [18]. This evaluation used a Box-Behnken experimental design (described in Section 2.4).

The metrics of *R^2^* and *RMSE* helped determine the optimum input parameters for the MLR model. Specifically, we observed the *R^2^* and *RMSE* as we added input variables to the model in the sequence of concentration, temperature, RH, and intercept (Figure 5 and Appendix A). Group *R^2^* was the best (*R^2^* = 0.878) when including concentration, temperature, and RH in the MLR model. *R^2^* dropped to 0.751 for the group after adding the intercept term. Individually, *R^2^* for each sensor was also the best when ignoring the intercept term. In addition, the *p*-value and 95% confidence interval (95% CI) of the intercept indicated that this intercept term was not significant (Appendix A). The group *RMSE* was similar with and without the intercept term (*RMSE* = 13.32 ppb, *RMSE* = 13.28, respectively). Consequently, the intercept term was ignored in the MLR model.

The resulting MLR relationship for the eight sensors was given by:(13)$$\hat{y}=0.552*c+0.399*T-0.142*RH$$
where $\hat{y}$ is the predicted sensor signal, c is the formaldehyde concentration, T is the temperature, and RH is the relative humidity. The equation indicates that sensor response is largely dependent on formaldehyde concentration (*p*-value = 1.01 × 10^−31^), positively related to temperature (*p*-value = 3.06 × 10^−12^) and negatively related to RH (*p*-value = 1.02 × 10^−7^). If the low-cost sensors are used to predict formaldehyde concentration, the MLR model could be converted into:(14)$$c=1.812*\hat{y}-0.723*T+0.257*RH$$

Temperature and RH can affect the baseline and sensitivity of electrochemical sensors [50,61]. Some previous studies have shown that temperature can possibly affect the electrodes of the electrochemical sensors [25] and RH may affect the humidity equilibrium between sample gas and the electrodes [24]. Over the long term, RH also changes the electrolyte chemistry, resulting in a change in zero current and sensitivity [33,34,36]. In addition, sensor type, sensor material, experimental setup and model selection could also affect the sensor response to different temperature and RH conditions.

Sensor response to temperature and RH

Alonso et al. [37] evaluated the effects of temperature and RH on the performance of a micro fuel cell formaldehyde sensor using three different models (OLS, ML, and REML). Their laboratory evaluation included a formaldehyde concentration range of 17.3–477.1 ppb (22–606 µg/m^3^), a temperature range of 19.9–27.7 °C and an RH range of 26.8–72.6%. Their OLS model indicated that both temperature and RH were significant factors for the sensor performance (temperature negatively related and RH positively related). However, the other two models in their study showed that RH was not a significant parameter; temperature significant and was positively related to sensor performance.

Chattopadhyay et al. [26] compared the performance of three different formaldehyde sensors over the temperature range of 22–50 °C and the RH range of 8–85% by looking into the *MAE* of each sensor compared to a reference instrument (Gasera One Formaldehyde). Their study showed that both metal oxide sensors (SGP30, BME680) and electrochemical sensor (ZE08-CH2O) were sensitive to increasing temperature. Specifically, the metal oxide sensors exhibited larger *MAE*s when temperature was above 45 °C. The electrochemical sensor (ZE08-CH2O) appeared to be insensitive to RH and exhibited a consistently low measurement *MAE* (29.43 ppb) over the RH range (8–85%). The BME680 sensor showed low *MAE* (48.33 ppb) at higher humidity (RH% > 45%) but high *MAE* (> 500 ppb) when RH is below 40%. The SGP30 sensor exhibited high sensitivity to humidity with an increasing *MAE* as humidity increased. However, they did not quantitatively evaluate the relationship of temperature and RH on sensor response.

b.Effects of RH

Previous studies reported different effects of RH on sensor response. Descamps et al. [39] found that an increased RH from 0% to 72.5% reduced their fluorescence-based sensor signal for indoor formaldehyde measurements. Chen et al. [60] reported that a Ga-doped ZnO sensor with Ag paste and Pt wires was not significantly affected by RH (0 to 70 ± 10%). Alolaywi et al. [38] studied the influence of RH on a MoOx/carbon nanocomposite-based electrochemical sensor at room temperature and found that increasing RH from 65% to 90% led to an increased signal response when exposed to 1 ppm formaldehyde gas.

### 3.3. Preliminary Cross-Sensitivity Tests

Appendix A summarizes the preliminary cross-sensitivity results. It shows a decrease in sensor signal when exposed to CO, NO, or NO_2_, compared to the average sensor signal when exposed to zero air. This decrease was statistically significant (*t*-test, *p*-value < 0.01). Isobutylene at a concentration of 100 ppb did not appear to affect the sensor signal (*t*-test, *p*-value > 0.06). In general, previous studies have reported cross sensitivity of electrochemical sensors to common air pollutants, including ozone (O_3_) cross sensitivity to NO_2_, and CO cross sensitivity to NO and NO_2_ [50]. Previous studies have also reported laboratory cross-sensitivity results for different types of formaldehyde sensors. Chang et al. [12] reported responses to ethanol, water vapor, oxygen and acetone that were 7 times lower than to formaldehyde for their functionalized phosphomolybdic acid (PMA)-based sensor. Li et al. [59] found that their hollow TiO_2_ microsphere sensor had a 10 times greater response to formaldehyde compared to methylbenzene, methanol, ethanol, acetone, and ammonia (3 ppm of formaldehyde and 3 ppm of potential interferent gas, at a temperature of 20 °C and an RH of 40%). Gautam et al. [28] reported that their Si-chip assisted MOS/SiNWs nanocomposite-based sensor had double the response to formaldehyde, compared to methanol, ethanol and acetone (1 ppm of interferent gas and 1 ppm of formaldehyde at room temperature). In summary, our study suggests that the SFA30^®^ is also sensitive to CO, NO and NO_2_. However, additional study is needed to further understand this sensor’s selectivity to formaldehyde in the presence of other relevant atmospheric trace gases.

### 3.4. Limitations

This study has several limitations. First, the experimental setup did not permit the study of sensor performance when temperature was below 0 °C or RH exceeded 77%. Second, we performed preliminary cross-sensitivity tests that included only a limited number of species (CO, NO, NO_2_, and isobutylene) concentrations. This preliminary cross-sensitivity study did not evaluate e the magnitude of this cross sensitivity or other potentially relevant species or mixtures of species. In the ambient environment, formaldehyde is part of a complex mixture of traces gases. However, this is a first step at understanding sensor performance. More sophisticated models [15,26] might be developed to evaluate sensor performance in an ambient environment.

## 4. Conclusions

This study comprehensively evaluated eight low-cost electrochemical formaldehyde sensors over a formaldehyde concentration range of 0–76 ppb in the laboratory, with a high-accuracy reference instrument (BBCEAS). In general, the sensors exhibited a good linearity of response (*R^2^* > 0.98 individually and > 0.95 as a group), low *LOD* (2.86–9.73 ppb individually and 11.3 ± 2.07 ppb as a group), good accuracy (1.84–10.1 ppb *RMSE* and 2.9–16.8% *NRMSE* individually, 3.96 ± 0.33 ppb *RMSE* and 6.23 ± 0.33% *NRMSE* as a group), acceptable repeatability (1.36–6.41% *CV* individually), moderate inter-sensor variability (ICC = 0.551) and reasonably rapid response (131–439 s). The effects of temperature (target range of 0–40 °C) and RH (target range of 0–75%) were also systematically investigated with a Box-Behnken experimental design and an MLR model. The MLR model indicates that sensor response is largely dependent on formaldehyde concentration, positively related to temperature and negatively related to RH. Overall, this study is important for understanding the performance of low-cost electrochemical formaldehyde sensors. It also reveals the feasibility of applying the sensors to field measurement of formaldehyde in the future.

## Figures and Tables

**Figure 1 sensors-23-07444-f001:**
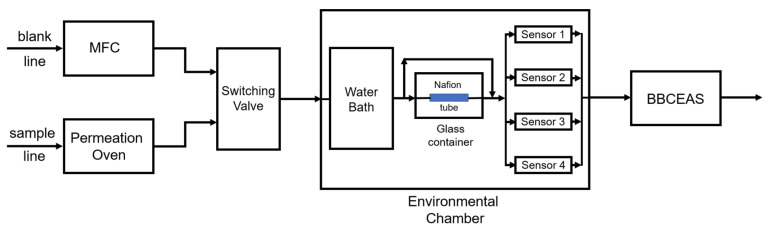
Flow chart of the laboratory evaluation system. All the tubing inside the system was PTFE tubing (OD 1/4″).

**Figure 2 sensors-23-07444-f002:**
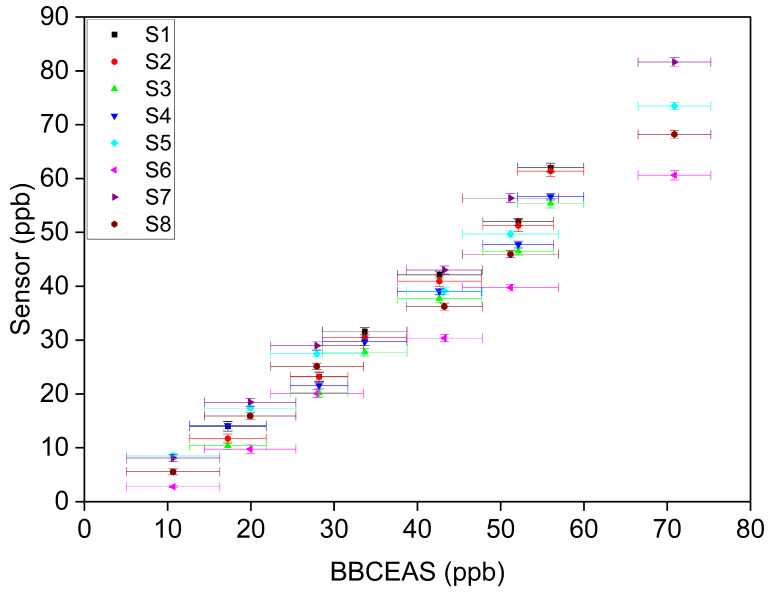
Response of eight sensors vs. the BBCEAS to formaldehyde concentrations ranging from 10.7 ppb to 70.9 ppb at 24 °C and approximately 15% RH. Error bars in X-axis denote the standard deviation of the BBCEAS measurement and error bars in Y-axis denote the standard deviation of the sensor response.

**Figure 3 sensors-23-07444-f003:**
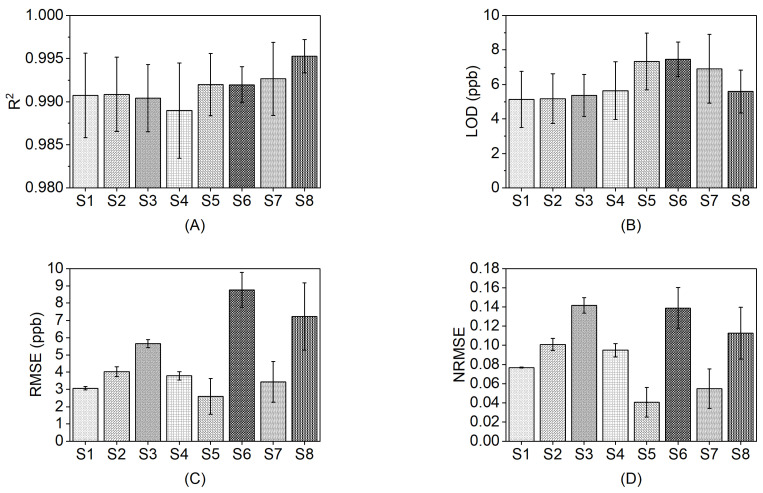
Performance metrics for each sensor in concentration-only tests at 24 °C and approximately 15% RH: (**A**) *R^2^*; (**B**) *LOD*; (**C**) *RMSE*; (**D**) *NRMSE*. Error bars denote the standard deviation for three repeated tests. The different patterns in the bars denote different sensors.

**Figure 4 sensors-23-07444-f004:**
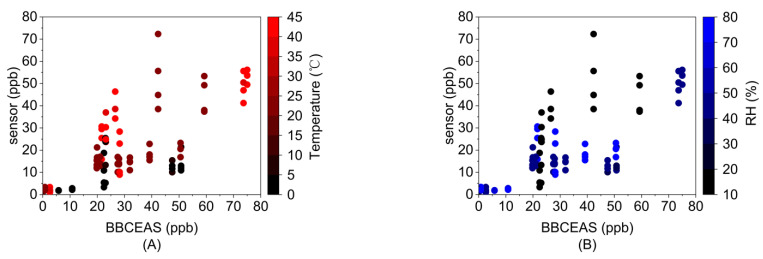
Sensor response vs. formaldehyde concentration, temperature, and RH: (**A**) Temperature (°C) and (**B**) RH (%). Experimental conditions selected by a Box-Behnken design. Note that this figure showed actual testing conditions. Target testing conditions can be found in Table 1.

**Figure 5 sensors-23-07444-f005:**
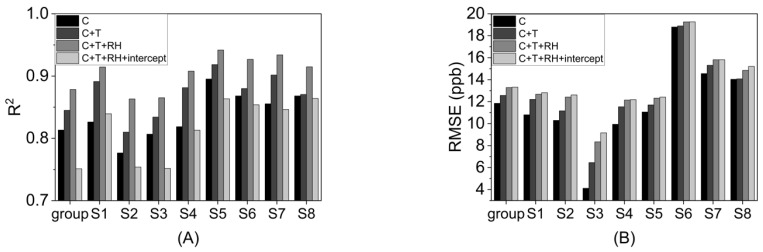
*R^2^* and *RMSE* for the MLR model variable selection: (**A**) *R^2^* and (**B**) *RMSE*. The following parameters were added sequentially: concentration (C), temperature (T), relative humidity (RH) and the intercept. Results were included for the group of all eight sensors as well as for the individual sensors.

**Table 1 sensors-23-07444-t001:** Box-Behnken design for environmental simulation tests of the sensors.

Test No.	Target Concentration * (ppb)	Target Temperature * (°C)	Target RH * (%)
1	25	0	15
2	25	40	15
3	25	0	75
4	25	40	75
5	0	0	45
6	0	40	45
7	50	0	45
8	50	40	45
9	0	20	15
10	0	20	75
11	50	20	15
12	50	20	75
13	25	20	45
14	25	20	45
15	25	20	45

* Note that these were target concentrations, temperatures, and RHs. Actual conditions can be found in Appendix A.

**Table 2 sensors-23-07444-t002:** Performance metrics of all sensors as a group in three repeated concentration-only tests.

Metrics	Average	Standard Deviation
*R^2^*	0.964	0.012
*LOD* (ppb)	11.3	2.07
*RMSE* (ppb)	3.96	0.33
*NRMSE*%	6.23	0.33

## Data Availability

The data presented in this study may be obtained from the authors upon reasonable request.

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
