# Peer review of "Laboratory Performance Evaluation of a Low-Cost Electrochemical Formaldehyde Sensor"

_sensors, 2023, doi:10.3390/s23177444_

Round 1
Reviewer 1 Report
Comparing the performance of electrochemical formaldehyde sensors is crucial for the development of highly sensitive formaldehyde detectors. This article has a certain innovative type and can be considered for publication. However, the following issues require the author to think deeply and revise in the article.
1. Although the author provided the chart of evaluation system, it is far from enough. It is recommended to provide an intuitive instrument detection flow chart or electronic picture? Only then will the results be convincing.
2. In fact, the author only compared the formaldehyde detection performance of 8 electrochemical formaldehyde sensors. The title gave the impression that the author prepared a low-cost sensor. The title should be revised. Such as “a comparative study…”
3. As for the outdoor environments, the composition of the ambient atmosphere (concentration of carbon dioxide, oxygen, etc.) is crucial to the performance of formaldehyde detection. Has the author investigated it?
4. Some related references can be cited: 10.1016/j.ccr.2021.214280;10.1039/D2TA04433A;10.1021/acs.langmuir.2c02111
5. The purpose/ innovation of this manuscript should be throughly considered in introduction section and discussion section.
no need.
Reviewer 2 Report
This study assessed eight low-cost electrochemical formaldehyde sensors in the laboratory, finding good linearity, low LOD, accuracy, repeatability, and response time. The study systematically investigated temperature and RH effects, highlighting the sensors' potential for field measurements of formaldehyde. While the current study lacks novelty, I believe it still holds value as a meticulous and methodical laboratory report. It offers comprehensive information about the target analyte. This report is beneficial for commercial sensor applications and could potentially serve as a valuable blueprint for future research endeavors. The following questions and comments may help improve the article:
1- Discuss the types of electrochemical reactions that underlie the detection of formaldehyde by electrochemical sensors.
2- When referring to the limitations of previous studies, could you elaborate on the specific implications of not having systematic investigations of temperature and relative humidity effects on sensor performance?
3- Consider adding a sentence that elaborates on the concept of "hazardous air pollutants" and their relevance in the context of formaldehyde exposure.
4- Include a sentence or two about the relevance of indoor and ambient formaldehyde levels to contextualize the concentration ranges studied in these investigations.
5- In the first sentences of the third paragraph of the introduction, refer to the new research (such as DOI 10.1088/1361-6528/acc6d7) that has tried to improve the sensitivity of resistive electrochemical gas sensors with different techniques.
6- What is the main reason for choosing a SFA30? And what was the reason for dividing the sensors into two sets of four?
7- Could you elaborate on any observed differences in the response patterns among the eight sensors?
8- Consider providing a brief description of the slope and intercept of the linear regression model that was applied.
9- Please review more literature on the effect of temperature and humidity on the sensing performance of gas sensors.
10- When discussing the limitation regarding the use of formaldehyde as the sole gas source, you could mention any known potential interfering gases that are common in the environment.
The text is well-written and conveys the information clearly.
Reviewer 3 Report
This is a well-written contribution, whose results might be very interesting for the community. It is well organized and the English level is satisfactory. I only suggest to better highlight the novelty compared to the existing literature. It would be also interesting for the readers knowing the performance compared to other technologies as microwave gas/liquid sensors, e.g., Cardillo, E.; Tavella, F.; Ampelli, C. Microstrip Copper Nanowires Antenna Array for Connected Microwave Liquid Sensors. Sensors 2023, 23, 3750. https://doi.org/10.3390/s23073750.
